# Distributed-Framework Basin Modeling System: III. Hydraulic Modeling System

Xiaoning Li [1,2], Chuanhai Wang [1,2], Gang Chen [1,2,*], Xing Fang [3], Pingnan Zhang [2] and Wenjuan Hua [2]

1   State Key Laboratory of Hydrology—Water Resources and Hydraulic Engineering, Hohai University, Nanjing 210098, China; xzl0938@hhu.edu.cn (X.L.); chwang@hhu.edu.cn (C.W.)
2   College of Hydrology and Water Resources, Hohai University, Nanjing 210098, China; pingnanzhang@hhu.edu.cn (P.Z.); huawenjuan0106@126.com (W.H.)
3   Department of Civil and Environmental Engineering, Auburn University, Auburn, AL 36849-5337, USA; xing.fang@auburn.edu
*   Correspondence: gangchen@hhu.edu.cn; Tel.: +86-139-1302-9378

**Abstract:** A distributed-framework basin modeling system (DFBMS) was developed to simulate the runoff generation and movement on a basin scale. This study is part of a series of papers on DFBMS that focuses on the hydraulic calculation methods in runoff concentration on underlying surfaces and flow movement in river networks and lakes. This paper introduces the distributed-framework river modeling system (DF-RMS) that is a professional modeling system for hydraulic modeling. The DF-RMS contains different hydrological feature units (HFUs) to simulate the runoff movement through a system of rivers, storage units, lakes, and hydraulic structures. The river network simulations were categorized into different types, including one-dimensional river branch, dendritic river network, loop river network, and intersecting river network. The DF-RMS was applied to the middle and downstream portions of the Huai River Plain in China using different HFUs for river networks and lakes. The simulation results showed great consistency with the observed data, which proves that DF-RMS is a reliable system to simulate the flow movement in river networks and lakes.

**Keywords:** basin modeling; distributed-framework model; river network simulation; numerical method; hydraulic calculation





## 1. Introduction

Floods are extreme phenomena that need to be accurately assessed for the protection of mankind's activities. Hence, mathematical tools focused on hydraulic simulations are designated as the most holistic approach to describe/simulate flood events and determine flood hazards [1]. Hydraulic modeling includes one-dimensional (1D), two-dimensional (2D), or three-dimensional (3D) approaches, regarding the number of dimensions in which the flow path is solved, as well as non-numerical models [2]. The current coupling of hydraulic models with geographic information systems (GIS) has facilitated the development and application of hydraulic models [3]. The use of remote sensing products, such as digital elevation models (DEMs), in hydraulic modeling, is also a commonly used modern approach [4]. In this study, we focused on the basic theories and concepts of hydraulic modeling in the distributed-framework basin modeling system (DFBMS). The runoff concentration and routing model based on DEM is an essential component of the DFBMS [5,6], which was introduced in the second paper in this series. In this paper, we introduce another professional model system of DFBMS [7–9]: distributed-framework river modeling system (DF-RMS). The DF-RMS models the hydraulics of runoff movement through rivers, lakes, reservoirs, and hydraulic engineering structures using a one-dimensional or two-dimensional approach [10]. Based on the concept of a hydrological feature unit (HFU) described in the first two series papers, the modeling/study area/region can involve a combination of different HFUs for flow movement types such as plain river HFU, lakes and

reservoir HFU, or hydraulic engineering structure HFUs. The DF-RMS contains different types of river network components (Figure 1), such as a single river, dendritic river network, island river network, and intersecting river network, that can be applied in complex river network systems.

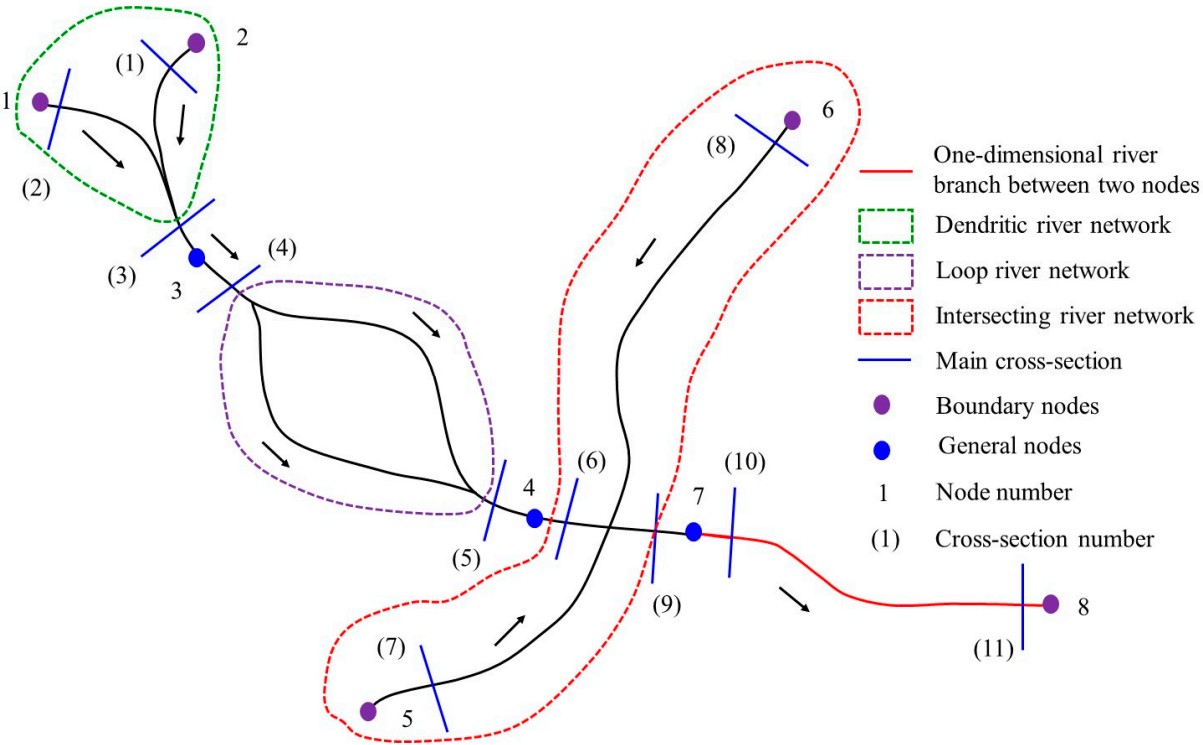

**Figure 1.** Different river network types and components in DF-RMS.

The finite difference method (FDM) and finite volume method (FVM) are commonly used to compute the flow movement in river networks and lakes in the previous study. The FDM includes explicit and implicit methods such as the alternating direction implicit (ADI) method [11,12] and the split-operator method [13]. Namiki [11] proposed a finite difference time domain (FDTD) algorithm to eliminate the Courant condition restraint. The algorithm is based on an alternating direction implicit method. The algorithm is stable both analytically and numerically, even when the Courant condition is not satisfied. The newly developed FDTD algorithm is more efficient than conventional FDTD schemes where the minimum cell size in the computational domain is required to be much smaller than the wavelength. Gustafsson [12] developed an alternating direction implicit difference scheme for solving shallow water equations. Gustafsson's scheme proved to be unconditionally stable for solving the linearized flow equations. Different iteration methods were developed for solving nonlinear algebraic equations including a quasi-Newton's method. The different iteration methods were tested numerically and compared for arbitrarily large time steps. Carfora [13] developed a semi-implicit, semi-Lagrangian, mixed finite difference-finite volume model for the spherical atmospheric shallow water equations. The main features are the vectorial treatment of the momentum equation and the finite volume approach for the continuity equation. Pressure and Coriolis terms in the momentum equation and velocity in the continuity equation are treated semi-implicitly. A split-operator method was introduced to preserve the symmetry of the numerical scheme.

The explicit methods are easy to adopt and code, while the computation time step is limited to the Courant condition. A polarization effect [14,15] will occur when the topography changes dramatically, and the Courant number will be larger than five for the explicit method. For the river networks, the width is always far smaller than the length. Therefore, a chasing method [14] could be used to solve the matrix for river

network flow [16]. The split-operator method is suited to vast flow movement areas such as lakes and simplified the procedures of lake flow movement simulation as an explicit method [16]. In this study, the boundary-fitted coordinate method was adapted to overcome the difficulties resulting from the natural stream's complicated layout (or stream orders) and the great disparity between length and width. Therefore, the irregular domain on the physical plane could be transferred into a rectangular computation domain. The splitting operator method and matrix chasing method was adapted to deal with the two-dimensional river flow as the different one-dimensional problems which improved the computation efficiency. The distributed-framework river modeling system was used to simulate a real case to test the system's computation efficiency and accuracy.

## 2. Materials and Methods

### 2.1. Plain River HFU

The runoff generated on the underlying surface was assigned differently to the river network in the hill area and plain area based on the digital elevation model. For the river network in the hill area, the runoff generated on the underlying surface was concentrated at the outlet of the sub-watershed and assigned to be upstream of the river. In the plain area, the land-use types for underlying surfaces included water, rain-fed land, construction land, and paddy field. The runoff generation process of different land-use types was calculated separately and lumped together for the sub-watershed based on the percentages of different land-use areas. As introduced in the second paper in the series, the fastest runoff concentration path (FRCP) with D8 procedure was promoted and adopted to deal with depressed pits on surface DEM and runoff path generation. In the D8 procedure, the runoff concentration path for a point was computed as the direction to the neighbor (based on 8-connectivity) which had minimum elevation and which was lower than the central point. The runoff on the underlying surface under grid processing was assigned to the nearest river cross-section individually rather than concentrated to the sub-watershed outlet or the upstream entrance. In the plain area, the runoff on each grid that was assigned to the corresponding river's cross-section could be determined based on the DEM with the FRCP method. In the FRCP method, it was assumed that the runoff would be concentrated along the path with the shortest time between the outlet and upstream cells in the underlying surface. The detailed information about FRCP can be found in the second paper in this series.

#### 2.1.1. River Network Component Generalization

Different types of river network components were considered in the DF-RMS (Figure 1), including single river reaches (the red solid line), dendritic river networks (connecting through a junction, shown inside green dotted lines), loop river networks (such as flow around an island, shown inside purple dotted lines), and intersecting river networks (inside red dotted lines). Figure 1 shows a combination of different river network types modeled by DF-RMS. The general nodes inside the river network were used to connect different components through the adjacent cross-section. Similarly, the boundary nodes were set to connect with boundary conditions such as a time series of water surface elevation or discharge.

#### 2.1.2. One-Dimensional River Flow Simulation

In this case, the hydrodynamic method was used to determine the water surface elevation and flow distribution along cross-sections for river networks in plain areas. One-dimensional Saint–Venant equations were used to describe the flow in the rivers:

$$\begin{cases} B\dfrac{\partial z}{\partial t} + \dfrac{\partial Q}{\partial x} = q \\ \dfrac{\partial Q}{\partial t} + \dfrac{\partial}{\partial x}\left(\dfrac{\alpha Q^2}{A}\right) + gA\dfrac{\partial z}{\partial x} + gA\dfrac{|Q|Q}{K^2} = qV_x \end{cases} \tag{1}$$

where $q$ is the lateral inflow (m$^2$/s) from the surrounding surfaces; $Q$ is the flow rate of the cross-section (m$^3$/s); $A$ is the flow area (m$^2$); $B$ is the flow width (m); $z$ is the water depth (m); $V_x$ is the velocity of lateral flow along the main river, which was zero in this study; $K$ is the conveyance coefficient, which indicates the actual river convey capacity; $\alpha$ is the momentum correction coefficient, which describes the velocity distribution along the cross-section. The momentum correction coefficient can be calculated with $\alpha = \dfrac{A}{K^2} \sum_{j=1}^{m} \dfrac{K_j^2}{A_j}$ when the friction slopes are the same for the main channel and overbank area; $m$ is the number of the main-channel and overbank-area regions; $A_j$ and $K_j$ are the flow area and conveyance of the $j$th flow region; $A$ and $K$ are the sum of $A_j$ and $K_j$, respectively. $\alpha = 1$ when the flow is limited in the main channel.

To discretize Equation (1) with a four-point linear implicit difference method for the $i$th and $(i + 1)$th cross-section of the river will yield Equation (2):

$$\begin{cases} -Q_i^{j+1} + Q_{i+1}^{j+1} + C_i z_i^{j+1} + C_i z_{i+1}^{j+1} = D_i \\ E_i Q_i^{j+1} + G_i Q_{i+1}^{j+1} - F_i z_i^{j+1} + F_i z_{i+1}^{j+1} = \Phi_i \end{cases} \tag{2}$$

Figure 2 shows the discretization of a one-dimensional river branch between the starting and ending cross-sections. Equation (2) was discretized as shown in Equation (3) and solved by the chasing method. The relationship between the flow rate and water depth at the starting and ending cross-section could be derived by the chasing method. Therefore, the flowrate could be expressed as the linear relationship between the water depth at the starting and ending cross-sections.

$$\begin{cases} Q_{L1} = \alpha + \beta z_{(I)} + \gamma z_{(J)} \\ Q_{L2} = \theta + \delta z_{(J)} + \mu z_{(I)} \end{cases} \tag{3}$$

where $Q_{L1}$ and $z_{(I)}$ are the flow rate and water depth, respectively, at the starting cross-section; $Q_{L2}$ and $z_{(J)}$ are the flow rate and water depth, respectively, at the ending cross-section. From the ending cross-section to the starting cross-section, the flow rate at every cross-section could be expressed as a linear relationship between the water depth at the current cross-section and the ending cross-section as shown as Equation (4):

$$Q_i = \alpha_i + \beta_i z_i + \gamma_i z_{(J)} \tag{4}$$

where $i = L_2 - 1, L_2 - 2, L_2 - 3, \ldots, L_1$.

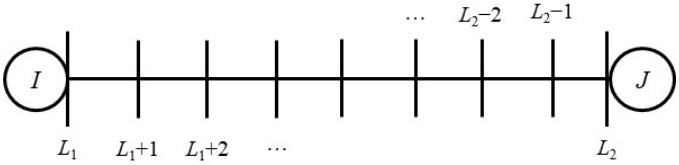

**Figure 2.** River discretization between the starting and ending cross-sections.

Similarly, from the starting cross-section to the ending cross-section, the flow rate at every cross-section could be expressed as the linear relationship between the water depth at the current cross-section and the starting cross-section as shown in Equation (5):

$$Q_i = \theta_i + \delta_i z_i + \mu_i z_{(I)} \tag{5}$$

where $i = L_1 + 1, L_1 + 2, L_1 + 3, \ldots, L_2$.

Once the water depth at the starting and ending cross-sections was given (boundary condition), Equations (4) and (5) could be solved for the $i$th cross-section between the starting and ending cross-sections:

$$\begin{cases} Q_i = \alpha_i + \beta_i z_i + \gamma_i z_{(J)} \\ Q_i = \theta_i + \delta_i z_i + \mu_i z_{(I)} \end{cases} \tag{6}$$

The water depth at the $i$th cross-section could be determined by solving Equation (6), yielding to:

$$z_i = \frac{\theta_i - \alpha_i + \mu_i z_{(I)} - \gamma_i z_{(J)}}{\delta_i - \beta_i} \tag{7}$$

The flow rate at the ith cross-section could be determined by substituting $z_i$ into Equation (6).

The intersection of two rivers was regarded as a computational node and treated via two different methods: (1) a node with a large surface area whose ponding volume could be calculated based on the surface area and ponding depth; (2) a node with a small surface area whose ponding volume could be neglected when the water depth changed in the node. The water depth in the node and the first cross-section were assumed to be the same, and the mass balance equation (Equation (8)) was used to calculate the water depth at the node:

$$\sum Q = A(z) \frac{\partial z}{\partial t} \tag{8}$$

where $A(z)$ is the ponding surface area in different water depths and $\sum Q$ is the inflow and outflow of the node.

In one-dimensional river flow simulations, computing the water depth at the node is the key step for the river network flow simulation. The flow rate and water depth could be determined after the water depth at the node was calculated. The water depth at the node was also a key parameter to couple with the runoff generation module. A mass balance matrix could be constructed for all nodes in the river network and solved with the boundary node's input value.

### 2.1.3. Two-Dimensional River Flow Simulation

The key step to couple one-dimensional and two-dimensional river network simulations is to find the water surface elevation at the node, which connects the one-dimensional model to a two-dimensional model. Different from the one-dimensional model, the computational nodes in a two-dimensional river network simulation were mainly selected from the smooth and steady section of the river branch. In the one-dimensional river network simulation, the nodes were primarily selected at the connection points of different river branches. Equation (9) is the governing equation that describes two-dimensional nonconservative flow motion in river networks:

$$\begin{cases} \dfrac{\partial z}{\partial t} + \dfrac{\partial hu}{\partial x} + \dfrac{\partial hv}{\partial y} = q \\[2mm] \dfrac{\partial u}{\partial t} + u\dfrac{\partial u}{\partial x} + v\dfrac{\partial u}{\partial y} + g\dfrac{\partial z}{\partial x} + g\dfrac{n^2\sqrt{u^2+v^2}}{h^{4/3}}u - fv = \dfrac{\partial}{\partial x}\left(E_x\dfrac{\partial u}{\partial x}\right) + \dfrac{\partial}{\partial y}\left(E_y\dfrac{\partial u}{\partial y}\right) \\[2mm] \dfrac{\partial v}{\partial t} + u\dfrac{\partial v}{\partial x} + v\dfrac{\partial v}{\partial y} + g\dfrac{\partial z}{\partial y} + g\dfrac{n^2\sqrt{u^2+v^2}}{h^{4/3}}v + fu = \dfrac{\partial}{\partial x}\left(E_x\dfrac{\partial v}{\partial x}\right) + \dfrac{\partial}{\partial y}\left(E_y\dfrac{\partial v}{\partial y}\right) \end{cases} \tag{9}$$

where $t$, $x$, and $y$ are time, $x$- and $y$-coordinates, respectively; $h$ and $z$ are the water depth and water surface elevation in the cell, respectively; $u$ and $v$ are the velocity in the $x$- and $y$-direction, respectively; $n$ and $f$ are the Manning's coefficient and Coriolis coefficient, respectively; $E_x$ and $E_y$ are the dispersion coefficient in the $x$- and $y$-direction, respectively; and $q$ is the source and sink term in the continuity equation that includes rainfall contribution, inflow, and outflow.

The boundary-fitted orthogonal coordinate transformation method was used to simulate complex boundaries and set the cell sizes when gridding the simulation river networks. The curvilinear grid system, with the computational boundary being coincident with the

real river network boundary, was numerically obtained by solving the Poisson equation. Grids in the boundary-fitted coordinate system were made up of two groups of lines, $\xi(x, y) = $ constant and $\eta(x, y) = $ constant, respectively. Each point $(x, y)$ in the physical domain corresponded with $(\xi, \eta)$ in the boundary-fitted coordinate system. The boundary in the physical domain coincided with the isoline of $\xi$ or $\eta$. Although the physical domain may be irregularly shaped, the transformed computational domain was foursquare. To simulate the irregular physical boundaries by the boundary-fitted coordinate transformation method, it was necessary to transform the basic differential equations and boundary conditions from $(x, y)$ space to a boundary-fitted coordinate system in $(\xi, \eta)$ space. In the DF-RMS, the grids were created by solving the Poisson equation, which means the transformations meet the Poisson equation (Equation (10)):

$$
\begin{cases}
\dfrac{\partial^2 \xi}{\partial x^2} + \dfrac{\partial^2 \xi}{\partial y^2} = P(\xi, \eta) \\[4mm]
\dfrac{\partial^2 \eta}{\partial x^2} + \dfrac{\partial^2 \eta}{\partial y^2} = Q(\xi, \eta)
\end{cases}
\tag{10}
$$

where $P$ and $Q$ are control functions that control the cell size and distribution.

With boundary-fitted orthogonal coordinate transformation, Equation (9) was transformed to Equation (11) in the $(\xi, \eta)$ space:

$$
\frac{\partial z}{\partial t} + \frac{1}{J}\Big[\frac{\partial\left(g_\eta u_* h\right)}{\partial \xi} + \frac{\partial\left(g_\xi v_* h\right)}{\partial \eta}\Big] = 0
\tag{11}
$$

$$
\frac{\partial u_*}{\partial t} + \frac{u_*}{g_\xi}\frac{\partial u_*}{\partial \xi} + \frac{v_*}{g_\eta}\frac{\partial u_*}{\partial \eta} + \frac{u_* v_*}{J}\frac{\partial g_\xi}{\partial \eta} - \frac{v_*^2}{J}\frac{\partial g_\eta}{\partial \xi} + \frac{gn^2 u_*\sqrt{u_*^2 + v_*^2}}{h^{\frac{4}{3}}} - fv_* + \frac{g}{g_\xi}\frac{\partial z}{\partial \xi}
$$
$$
= \frac{1}{g_\xi}\frac{\partial}{\partial \xi}\left(E_\xi A\right) - \frac{1}{g_\eta}\frac{\partial}{\partial \eta}\left(E_\eta B\right)
\tag{12}
$$

$$
\frac{\partial v_*}{\partial t} + \frac{u_*}{g_\xi}\frac{\partial v_*}{\partial \xi} + \frac{v_*}{g_\eta}\frac{\partial v_*}{\partial \eta} + \frac{u_* v_*}{J}\frac{\partial g_\eta}{\partial \xi} - \frac{u_*^2}{J}\frac{\partial g_\xi}{\partial \eta} + \frac{gn^2 v_*\sqrt{u_*^2 + v_*^2}}{h^{\frac{4}{3}}} + fu_* + \frac{g}{g_\eta}\frac{\partial z}{\partial \eta}
$$
$$
= \frac{1}{g_\eta}\frac{\partial}{\partial \eta}\left(E_\xi A\right) + \frac{1}{g_\xi}\frac{\partial}{\partial \xi}\left(E_\eta B\right)
\tag{13}
$$

where $A = \dfrac{1}{J}\left[\dfrac{\partial}{\partial \xi}\left(u_* g_\eta\right) + \dfrac{\partial}{\partial \eta}\left(v_* g_\xi\right)\right]$ and $B = \dfrac{1}{J}[\dfrac{\partial}{\partial \xi}\left(v_* g_\eta\right) - \dfrac{\partial}{\partial \eta}\left(u_* g_\xi\right)]$; $u_*$ and $v_*$ are the velocities in the $\xi$- and $\eta$-directions: $u_* = \dfrac{1}{g_\xi}(u\dfrac{\partial x}{\partial \xi} + v\dfrac{\partial y}{\partial \xi})$; $v_* = \dfrac{1}{g_\eta}(u\dfrac{\partial x}{\partial \eta} + v\dfrac{\partial y}{\partial \eta})$; $g_\xi$ and $g_\eta$ are the length and width of the cell; $g_\xi = \sqrt{x_\xi^2 + y_\xi^2}$ and $g_\eta = \sqrt{x_\eta^2 + y_\eta^2}$; $J$ is the cell area; $J = g_\xi \times g_\eta$, $n$ and $f$ are the Manning's coefficient and Coriolis coefficient, respectively; and $E_\xi$ and $E_\eta$ are the dispersion coefficient in the $\xi-$ and $\eta$-directions, respectively.

Figure 3 shows the boundary-fitted transformation and discretization of the physical river simulation domain (Figure 3a) and the transformed numerical simulation domain (Figure 3b). For convenience, the variables $z$, $u$, and $v$ are hereafter used to stand for water depth at the cell center, flow velocity in the $x(\xi)$-direction, and flow velocity in the $y(\eta)$-direction in the $(x, y)$ and $(\xi, \eta)$ space. The variables of water depth and flow velocity are numbered together in the computation code. The total variable number in the simulation domain for water depth, flow velocity in the $\xi$-direction, and flow velocity in the $\eta$-direction is $(N + 1) \times M$, $N \times M$, and $(N + 1) \times (M + 1)$, respectively. For convenience, three matrices were developed for variables $z$, $u$, and $v$, which are listed in the following text. The variables with superscript "0" ($z^0$, $u^0$, $v^0$) and "1" ($z^1$, $u^1$, $v^1$) mean the parameter's value of the current time step ($t$) and next time step ($t + \Delta t$), respectively.

$$
Z_{2k+1} = [z_{2k+1,2}, z_{2k+1,4}, \ldots, z_{2k+1,2l}, \ldots, z_{2k+1,2M}]^T \quad k = 0, 1, 2, \ldots, N
$$

$$U_{2k} = [u_{2k,2},\ u_{2k,4},\ \dots,\ u_{2k,2l},\ \dots, u_{2k,2M}]^T \quad k = 1,\ 2,\ \dots,\ N$$

$$V_{2k} = [v_{2k+1,3},\ v_{2k+1,5},\ \dots,\ u_{2k,2l+1},\ \dots, u_{2k,2M-1}]^T \quad k = 0,\ 1,\ 2,\ \dots,\ N$$

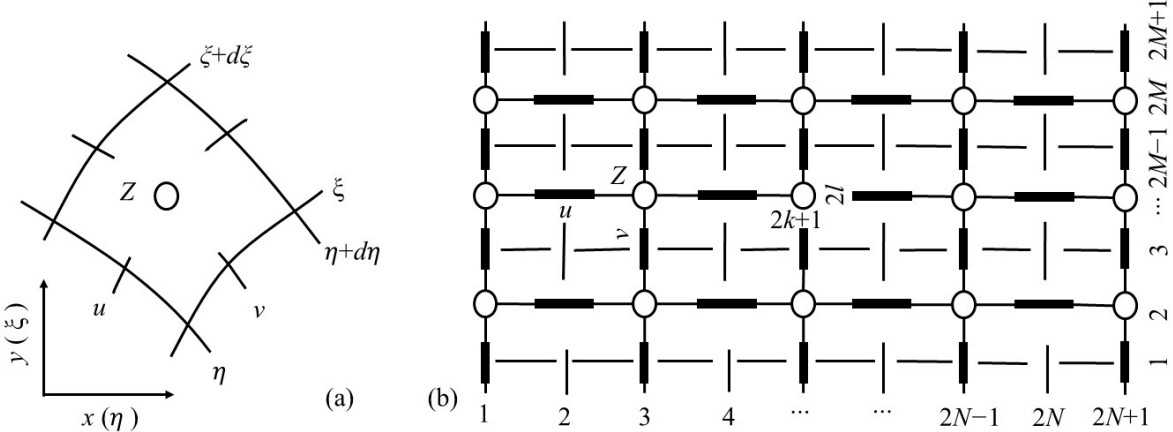

**Figure 3.** Boundary-fitted transformation and discretization of $(x,\ y)$ (**a**) and $(\xi,\ \eta)$ (**b**) space.

Two boundary conditions of riverbank could be selected in the model: (1) wall condition, which means $v = 0$ and $\dfrac{\partial u}{\partial \xi} = 0$ at the boundary; (2) inflow or outflow, which means $v$ = inflow or outflow velocity and $\dfrac{\partial u}{\partial \xi} = 0$ at the boundary. The water depth of upstream and downstream are given as boundary conditions in the simulation.

(1)   The discretization of continuity Equation (Equation (11))

To discretize the continuity equation (Equation (11)) at node $(2k+1,\ 2l)$ for $\dfrac{\partial z}{\partial t}$ will yield:

$$\frac{\partial z}{\partial t} = \frac{z^1_{2k+1,2l} - z^0_{2k+1,2l}}{\Delta t}$$

For the nonlinear term, $g_\eta hu$ and $g_\eta hv$, which yield:

$$\begin{cases} g_\eta hu = g_\eta h^0 u^1 + g_\eta u^0 z^1 - g_\eta u^0 z^0 \\ g_\xi hv = g_\xi h^0 v^1 + g_\xi v^0 z^1 - g_\xi v^0 z^0 \end{cases}$$

In this case, the nonlinear term will be discretized as:

$$\frac{\partial(g_\eta hu)}{\partial \xi} = \frac{(g_\eta h^0 u^1 + g_\eta u^0 z^1 - g_\eta u^0 z^0)_{2k+2,2l} - (g_\eta h^0 u^1 + g_\eta u^0 z^1 - g_\eta u^0 z^0)_{2k,2l}}{\Delta \xi}$$

and

$$\frac{\partial(g_\xi hv)}{\partial \eta} = \frac{(g_\xi h^0 v^1 + g_\xi v^0 z^1 - g_\xi v^0 z^0)_{2k+1,2l+1} - (g_\xi h^0 v^1 + g_\xi v^0 z^1 - g_\xi v^0 z^0)_{2k+1,2l-1}}{\Delta \eta}$$

where $z_{2k,2l}$ is the water surface elevation of the node $(2k, 2l)$, $z_{2k,2l} = (z_{2k-1,2l} + z_{2k+1,2l})/2$; $h_{2k,2l}$ is the water depth of node $(2k, 2l)$ and equal to the water surface elevation minus cell bottom elevation; $z_{2k+1,2l+1}$ is the water surface elevation of the node $(2k+1, 2l+1)$, $z_{2k+1,2l+1} = (z_{2k+1,2l} + z_{2k+1,2l+2})/2$.

Finally, the continuity equation was discretized as in the following formation:

$$\alpha_1 z^1_{2k-1,2l} + \alpha_2 z^1_{2k+1,2l-2} + \alpha_3 z^1_{2k+1,2l} + \alpha_4 z^1_{2k+1,2l+2} + \alpha_5 z^1_{2k+3,2l} + \beta_1 u^1_{2k,2l} +$$
$$\beta_2 u^1_{2k+2,2l} + \gamma_1 v^1_{2k+1,2l} + \gamma_2 v^1_{2k+1,2l+1} = \Phi \qquad (14)$$
$$k = 1,\ 2,\ 3,\ \dots,\ N-1; l = 1,\ 2,\ 3,\ \dots,\ M$$

In Equation (14):

$\alpha_1 = -\left(g_\eta u^0\right)_{2k,2l}/2$

$\alpha_2 = -\left(g_\xi v^0\right)_{2k+1,2l-1}/2$

$\alpha_3 = \dfrac{J_{2k+1,2l}}{\Delta t} + \left[\left(g_\eta u^0\right)_{2k+2,2l} - \left(g_\eta u^0\right)_{2k,2l} + \left(g_\xi v^0\right)_{2k+1,2l+1} - \left(g_\xi v^0\right)_{2k+1,2l-1}\right]/2$

$\alpha_4 = \left(g_\eta v^0\right)_{2k+1,2l+1}/2$

$\alpha_5 = \left(g_\eta u^0\right)_{2k+2,2l}/2$

$\beta_1 = -\left(g_\eta h^0\right)_{2k,2l}$

$\beta_2 = \left(g_\eta h^0\right)_{2k+2,2l}$

$\gamma_1 = -\left(g_\xi h^0\right)_{2k+1,2l-1}$

$\gamma_2 = \left(g_\xi h^0\right)_{2k+1,2l+1}$

$\Phi = \dfrac{J_{2k+,2l} z^0_{2k+1,2l}}{\Delta t} + \left[\left(g_\eta u^0 z^0\right)_{2k+2,2l} - \left(g_\eta u^0 z^0\right)_{2k,2l} + \left(g_\xi v^0 z^0\right)_{2k+1,2l+1} - \right.$
$\left(g_\xi v^0 z^0\right)_{2k+1,2l-1}\right]/2.$

The matrix was built-up with the following format:

$$A1_k Z_{2k-1} + B1_k Z_{2k+1} + C1_k Z_{2k+3} + D1_k V_{2k+1} + E1_k U_{2k} + F1_k U_{2k+2} = H1_k \qquad (15)$$
$$k = 1,\ 2,\ 3,\ \dots,\ N-1$$

where the dimensions of matrices $A1_k$, $B1_k$, $C1_k$, $E1_k$, and $F1_k$ are $M \times M$; the dimensions of the matrix $D1_k$ are $M \times (M-1)$; and the dimension of the matrix $H1_k$ is $M$.

(2) The discretization of momentum Equation (Equation (12))

To discretize the momentum equation in the $\xi$-direction (Equation (12)) at node $(2k,\ 2l)$ will yield:

$$\frac{\partial u}{\partial t} = \frac{u^1_{2k,2l} - u^0_{2k,2l}}{\Delta t}$$

To discretize the terms $\dfrac{u}{g_\xi}\dfrac{\partial u}{\partial \xi}$ and $\dfrac{v}{g_\eta}\dfrac{\partial v}{\partial \eta}$ with the upwind scheme will yield:

$$\frac{u}{g_\xi}\frac{\partial u}{\partial \xi} = \frac{u^0}{g_\xi}\frac{\partial u}{\partial \xi} \qquad \frac{v}{g_\eta}\frac{\partial v}{\partial \eta} = \frac{v^0}{g_\eta}\frac{\partial v}{\partial \eta}$$

The term $\dfrac{g}{g_\xi}\dfrac{\partial z}{\partial \xi}$ was discretized as follows:

$$\frac{g}{g_\xi}\frac{\partial z}{\partial \xi} = \frac{g}{g_\xi}\frac{z^1_{2k+1,2l} - z^1_{2k-1,2l}}{\Delta \xi}$$

The term $\dfrac{uv}{J}\dfrac{\partial g_\xi}{\partial \eta} - \dfrac{v^2}{J}\dfrac{\partial g_\eta}{\partial \xi}$ was discretized as follows:



$$\frac{uv}{J}\frac{\partial g_\xi}{\partial \eta} - \frac{v^2}{J}\frac{\partial g_\eta}{\partial \xi} = (\frac{u^0}{J}\frac{\partial g_\xi}{\partial \eta}$$

$$- \frac{v^0}{J}\frac{\partial g_\eta}{\partial \xi}) \times (\frac{v^1_{2k-1,2l-1} + v^1_{2k-1,2l+1} + v^1_{2k+1,2l-1} + v^1_{2k+1,2l+1}}{4})$$

The term $\dfrac{gn^2 u\sqrt{u^2+v^2}}{h^{\frac{4}{3}}}$ was discretized as follows:

$$\frac{gn^2 u\sqrt{u^2+v^2}}{h^{\frac{4}{3}}} = g\frac{n^2\sqrt{(u^0)^2 + (v^0)^2}}{(h^0)^{4/3}}u^1$$

Substituting all discretized terms into Equation (12) and building-up the linear matrix for the node $(2k, 2l)$ yields:

$$A2_k Z_{2k-1} + B2_k Z_{2k+1} + C2_k U_{2k-2} + D2_k U_{2k} + E2_k U_{2k+2} + F2_k V_{2k-1} + G2_k V_{2k+1} = H2_k$$
$$k = 1, 2, 3, \ldots, N \tag{16}$$

Similarly, we can discretize the momentum equation in the $\eta$-direction (Equation (21)) at node $(2k+1, 2l+1)$ and build-up the linear matrix in the following format:

$$A3_k Z_{2k+1} + B3_k U_{2k} + C3_k U_{2k+2} + D3_k V_{2k-1} + E3_k V_{2k+1} + F3_k V_{2k+3} = H3_k$$
$$k = 1, 2, 3, \ldots, N-1 \tag{17}$$

where the dimensions of matrices $A2_k$, $B2_k$, $C2_k$, $D2_k$, and $E2_k$ are $M \times M$; the dimensions of matrices $F2_k$ and $G2_k$ are $M \times (M-1)$; the dimensions of matrices $A3_k$, $B3_k$, and $C3_k$ are $(M-1) \times M$; the dimensions of matrices $D3_k$, $E3_k$, and $F3_k$ are $(M-1) \times (M-1)$; and the dimensions of matrices $H2_k$ and $H3_k$ are $M$ and $M-1$, respectively.

(3)  Solving the continuity and momentum matrix with boundary condition

The $u$, $v$, and $z$ variables could be solved by combining Equations (15)–(17) and the known boundary variables with the chasing method.

For the upstream boundary condition, we have $z_1 = Z_1(t)$ and $v_1 = 0$ to plot into Equation (16), which is in the following format:

$$A2_1 Z_1 + B2_1 Z_3 + D2_1 U_2 + E2_1 U_4 + G2_1 V_3 = H2_1$$

We can rearrange the equation to develop the relationship between these unknown variables:

$$U_2 = (UZ_1 \cdot Z_3) + (UU_1 \cdot U_4) + (UV_1 \cdot V_3) + UF_1 \tag{18}$$

We plotted two upstream boundary condition variables and Equation (18) into Equation (17) to find $V_{2k+1}$:

$$V_{2k+1} = (VZ_k \cdot Z_{2k+1}) + (VU_k \cdot U_{2k+2}) + (VV_k \cdot V_{2k+3}) + VF_k$$
$$k = 1, 2, 3, \ldots, N-1 \tag{19}$$

We plotted upstream boundary condition $z_1 = Z_1(t)$, substituting Equations (18) and (19) into Equation (15) to find $Z_{2k+1}$:

$$Z_{2k+1} = (ZZ_k \cdot Z_{2k+3}) + (ZU_k \cdot U_{2k+2}) + (ZV_k \cdot V_{2k+3}) + ZF_k$$
$$k = 1, 2, 3, \ldots, N-1 \tag{20}$$

We substituted Equations (18)–(20) into Equation (16) to find $U_{2k+2}$:

$$U_{2k+2} = (UZ_k{\cdot}Z_{2k+3}) + (UU_k{\cdot}U_{2k+4}) + (UV_k{\cdot}V_{2k+3}) + UF_k$$
$$k = 1, 2, 3, \ldots, N-1 \tag{21}$$

For $k = N - 1$, $U_{2N}$ was determined by the following equation:

$$U_{2N} = (UZ_N{\cdot}Z_{2N+1}) + (UU_N{\cdot}U_{2N+2}) + (UV_N{\cdot}V_{2N+1}) + UF_N$$

where the dimensions of matrices $UZ_k$, $UU_k$, $ZZ_k$, and $ZU_k$ are $M \times M$; the dimensions of matrices $UV_k$ and $ZV_k$ are $M \times (M-1)$; the dimensions of matrices $VZ_k$ and $VU_k$ are $(M-1) \times M$; the dimensions of matrix $VV_k$ are $(M-1) \times (M-1)$; the dimension of matrices $UF_k$ and $ZF_k$ is $M$; and the dimension of the matrix $VF_k$ is $M-1$.

The downstream boundary conditions are also given as $z_{2N+1} = Z_{2N+1}(t)$ and $v_{2N+1} = 0$. It was assumed that $U_{2N+2} = U_{2N}$, so we can substitute these known variables into Equation (21) to find the variables $Z_{2N-1}, V_{2N-1}, U_{2N-2}, \ldots, U_2$ for the current time step. Three properties were found in the matrix buildup and chasing method procedure: (1) most of the matrix was composed of three diagonal matrices ($A1_k$, $B1_k$, $C1_k$, $D1_k$, $E1_k$, $F1_k$, $A2_k$, $B2_k$, $C2_k$, $D2_k$, $E2_k$, $F2_k$, $G2_k$, $A3_k$, $B3_k$, $C3_k$, $D3_k$, $E3_k$, and $F3_k$) and some other diagonal matrices, which reduces the computational load and cost to a considerable extent; (2) the ratio of river width to river length is tiny for most of the river network ($M \ll N$), which means the matrix operation was not very complex; (3) this method was mostly focused on the matrix operation, which is similar to the implicit method, and the computational time step could be huge with a Courant number ~100 and different to the explicit method.

### 2.1.4. Flow Simulation in River Intersections

Most previous studies treated the main river and tributary separately [17,18], which only transfers the data between the main river and tributary at the river intersection, rather than directly dealing with the river intersection in the algorithm. The alternating direction implicit ADI method [12,19] was used to solve the shallow water equation so that the polarization effect [14,15] would occur in the intersection part when the Courant number was larger than five for the explicit method. The matrix for river intersection was built-up to find the variables with the chasing method. Details of the matrix buildup process can be found in the Supplementary Materials published with the paper about flow simulation in river intersections; the process is similar to the two-dimensional river flow simulation matrix buildup process. As shown in Figure 4, the water surface elevation and flow velocity in the $\xi$-direction is $Na + 1$ and the flow velocity in the $\eta$-direction is $Nb$. The computational node for the tributary in the first row was the same as the part of the nodes in the main river (the dotted circle in Figure 4), which coupled the matrix of the tributary with that of the main river. Different to the section between node $k = 1$ and $k = (LI - 1)$ of the main river, the connection between $k = (LI - 1)$ and $k = (LI - 2) + Mb$ of the main river and tributary was discretized in the following format:

$$A1_k Z_{2k-1} + B1_k Z_{2k+1} + C1_k Z_{2k+3} + D1_k V_{2k+1} + E1_k U_{2k} + F1_k U_{2k+2}$$
$$+ P1_k U_1^b + Q1_k Z_2^b + R1_k V_2^b = H1_k \tag{22}$$
$$k = LI - 2, LI - 1, \ldots, LI + Mb - 2$$

$$A2_k Z_{2k-1} + B2_k Z_{2k+1} + C2_k U_{2k-2} + D2_k U_{2k} + E2_k U_{2k+2} + F2_k V_{2k-1}$$
$$+ G2_k V_{2k+1} + P2_k U_1^b + Q2_k Z_2^b + R2_k V_2^b = H2_k \tag{23}$$
$$k = LI - 1, \ldots, LI + Mb - 2$$

$$A3_k Z_{2k+1} + B3_k U_{2k} + C3_k U_{2k+2} + D3_k V_{2k-1} + E3_k V_{2k+1} + F3_k V_{2k+3}$$
$$+ P3_k U_1^b + Q3_k Z_2^b + R3_k V_2^b = H3_k \tag{24}$$
$$k = LI - 2, LI - 1, \ldots, LI + Mb - 2$$

where the dimensions of matrices $P1_k$, $Q1_k$, $P2_k$, and $Q2_k$ are $M_1 \times M_2$; the dimensions of matrices $R1_k$ and $R2_k$ are $Ma \times (Mb - 1)$; the dimensions of matrices $P3_k$ and $Q3_k$ are $(Ma - 1) \times Mb$; the dimensions of matrix $R3_k$ are $(Ma - 1) \times (Mb - 1)$; the dimension of the matrices $UF_k$ and $ZF_k$ is $M$; and the dimension of the matrix $VF_k$ is $Mb - 1$.

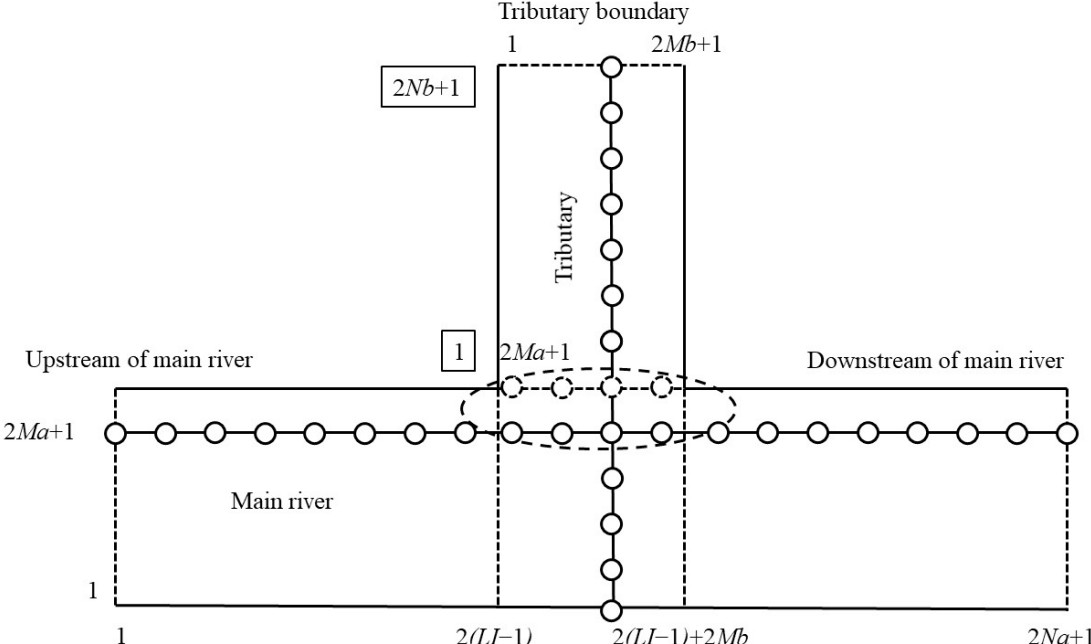

**Figure 4.** Detail of river intersection generalization.

### 2.1.5. Flow Simulation in a Loop River

Figure 5 shows the sketch of a loop river that includes the main river and a tributary. The start and end of the tributary coincide with two parts of the main river. More generally, it deals with the flow around an island: upstream flow split and downstream flow combination. The area of overlap was used to couple the main river and tributary when discretizing the loop river. Three steps were adopted when discretizing the whole loop river to make sure the system was orthogonal. (1) The boundary-fitted coordinate system method was used to discretize the entire loop river system; (2) we discretized only the main river and considered the overlap area as the boundary; (3) we discretized only the tributary and considered the overlap area as the boundary. Steps (2) and (3) may be repeated several times to make sure the whole system is orthogonal.

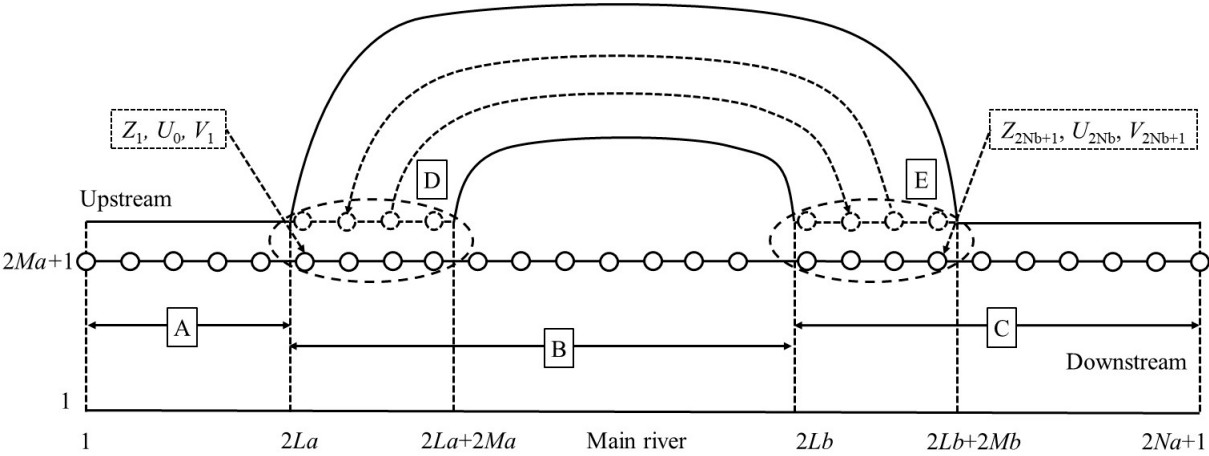

**Figure 5.** Details of loop river discretization generalization.

The key to build up the matrix is to find the linear relationship between unknown variables (water surface elevation and flow velocity) at different nodes and given variables at the upstream and downstream boundaries. Details of the matrix buildup process can be found in the Supplementary Materials (flow simulation in a loop river); the process is similar to the two-dimensional river flow simulation matrix buildup process.

### 2.1.6. Flow Simulation for a River Network

Taking the combination of different kinds of river network as an example (Figure 1), it is necessary to determine the water surface elevation at three boundary nodes (1, 2, and 3) as well as the relationship between the flow of cross-sections (1, 2, and 3) and the water surface elevation at boundary nodes to build the matrix. The flow and water surface elevation at sub-cross-sections of river sections 1–3 and 2–3 can be obtained by solving the matrix. However, the water surface elevation at node 3 and flow of cross-section (3) is often unknown in reality. Therefore, it is necessary to obtain the water surface elevation of node 3 first. Similarly, it is necessary to find the relationship between the water surface elevation at the boundary node (node 3 and 4) and the flow at the main cross-sections (4) and (5) to build the matrix for all variables of the loop river network. For the intersecting river network component, the relationship of the water surface elevation at boundary nodes (4, 5, 6, and 7) and the flow of main cross-sections (6, 7, 8, and 9) were necessary to build the matrix. Overall, the matrix of boundary and general nodes was first built to calculate the water surface elevation for every node in the river network. All these nodes can be named water surface elevation nodes. Then the flow and water surface elevation of sub-cross-sections can be achieved through the known node value.

### 2.2. Lakes and Reservoir's HFU

Lakes, reservoirs, and floodplains are considered to have the same kind of runoff storage components that provide the storage volume for the whole system. The flow movement in the storage component was mainly driven by the factors of inflow, outflow, wind, etc. Zero-dimensional and two-dimensional models were considered to simulate the flow movement in lake and reservoir HFUs. In DF-RMS, zero-dimensional was adopted to simulate the storage capacity of lakes, and a two-dimensional model was adopted to simulate the flow movement in lakes.

### 2.2.1. Zero-Dimensional Flow Simulation in Lakes

A zero-dimensional model was used to simulate the storage capacity of lakes rather than the flow movement. In the zero-dimensional model, the water surface elevation and storage area were used to solve the mass balance equation (Equation (25)):

$$\sum Q = A(Z)\frac{\partial Z}{\partial t} \tag{25}$$

where $A(Z)$ is the storage area at different water surface elevations; $\sum Q$ is the sum of inflow, outflow, and runoff generation.

Equation (25) was discretized in the following format to find the water surface elevation:

$$\sum Q = A(Z_0)\frac{Z^1 - Z^0}{\Delta t} \tag{26}$$

where $\Delta t$ is the computational time step and $Z^1$ and $Z^0$ are the water surface elevation of the next time level and the current time level, respectively.

### 2.2.2. Two-Dimensional Flow Simulation in Lakes

A shallow water equation (SWE) was used to simulate the flow movement driven by the wind, inflow, and outflow for lakes, flood plain, and reservoirs:

$$
\begin{cases}
\dfrac{\partial z}{\partial t} + \dfrac{\partial hu}{\partial x} + \dfrac{\partial hv}{\partial y} = q \\[2mm]
\dfrac{\partial hu}{\partial t} + u\dfrac{\partial hu}{\partial x} + v\dfrac{\partial hu}{\partial y} + gh\dfrac{\partial z}{\partial x} = -gu\dfrac{\sqrt{u^2 + v^2}}{c^2} + fhv + \dfrac{\rho_a c_w \sqrt{w_x^2 + w_y^2}}{\rho} w_x \\[2mm]
\dfrac{\partial hv}{\partial t} + u\dfrac{\partial hv}{\partial x} + v\dfrac{\partial hv}{\partial y} + gh\dfrac{\partial z}{\partial y} = -gv\dfrac{\sqrt{u^2 + v^2}}{c^2} - fhu + \dfrac{\rho_a c_w \sqrt{w_x^2 + w_y^2}}{\rho} w_y
\end{cases}
\tag{27}
$$

where $t$, $x$, and $y$ are the time, $x$-coordinate, and $y$-coordinate, respectively; $h$ and $z$ are the water depth and water surface elevation in the cell, respectively; $u$ and $v$ are the velocities in $x$- and $y$-direction, respectively; $c$ and $f$ are the Chezy coefficient and Coriolis coefficient, respectively; $\tau_{wx}$ and $\tau_{wy}$ are the wind stress in the $x$-direction and $y$-direction, respectively; $\rho$ and $\rho_a$ are the water density and air density, respectively; $c_w$ is the wind drag coefficient; $w_x$ and $w_y$ are the wind velocity in $x$-direction and $y$-direction, respectively; and $q$ is the source and sink term in the continuity equation that includes rainfall contribution, inflow, and outflow.

The split-operator approach is a popular algorithm in computational fluid dynamics: operator-splitting techniques have been widely utilized in atmospheric modeling studies [20] to decouple reactions from convection and diffusion or convection from diffusion. The split-operator method was also used to decompose the momentum equation of the incompressible Navier–Stokes equations to solve the linked pressure–velocity problem [21].

The split-operator approach was used to split the governing equation into two steps:

(1)  The first step:

$$
\begin{cases}
\dfrac{\partial z}{\partial t} = 0 \\[2mm]
\dfrac{\partial hu}{\partial t} + u\dfrac{\partial hu}{\partial x} + v\dfrac{\partial hu}{\partial y} = 0 \\[2mm]
\dfrac{\partial hv}{\partial t} + u\dfrac{\partial hv}{\partial x} + v\dfrac{\partial hv}{\partial y} = 0
\end{cases}
\tag{28}
$$

(2)  The second step:

$$
\begin{cases}
\dfrac{\partial z}{\partial t} + \dfrac{\partial hu}{\partial x} + \dfrac{\partial hv}{\partial y} = q \\[2mm]
\dfrac{\partial hu}{\partial t} + gh\dfrac{\partial z}{\partial x} = -gu\dfrac{\sqrt{u^2 + v^2}}{c^2} + fhv + \dfrac{\rho_a c_w \sqrt{w_x^2 + w_y^2}}{\rho} w_x \\[2mm]
\dfrac{\partial hv}{\partial t} + gh\dfrac{\partial z}{\partial y} = -gv\dfrac{\sqrt{u^2 + v^2}}{c^2} - fhu + \dfrac{\rho_a c_w \sqrt{w_x^2 + w_y^2}}{\rho} w_y
\end{cases}
\tag{29}
$$

The finite volume method was used to solve the equations under an uneven rectangular grid cell coordinate. As shown in Figure 6, the flux between cell $i$ and cell $j$ could be calculated using the following discretized equation:

$$
\frac{q_x - q_x^0}{\Delta t} + gh^0 \frac{z_j - z_i}{\Delta x} + g\frac{|V|}{c^2(h^0)^2} q_x - fq_y - \frac{1}{\rho}\tau_{wx}
\tag{30}
$$

where $q_x$ and $q_x^0$ are the flow discharge per meter of the next time level and the current time level in the $x$-direction, respectively; $h^0$, $z_i$ and $z_j$ are the water depth and water surface elevation of cell $i$ and $j$; $|V|$ is a variable related to $q_x$ and $q_y$, $|V| = \sqrt{q_x^2 + q_y^2}$; $\tau_{wx}$ is the wind stress in the $x$-direction; and $q_y$ is the flow discharge per meter in the $y$-direction.

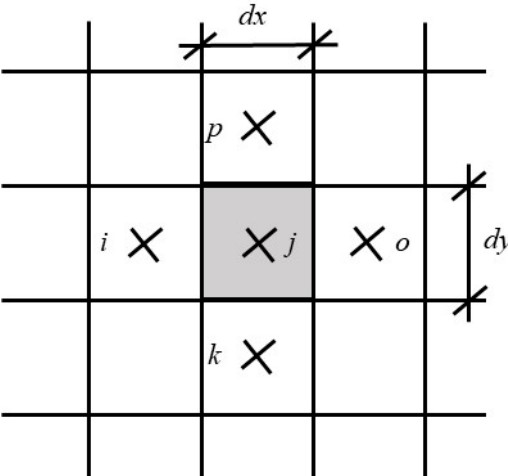

**Figure 6.** Discretization of the 2D lake simulation domain.

Finally, the equation was simplified and reorganized in the following format to calculate the unit width flow from cell $i$ to cell $j$:

$$q_x = \delta_0(z_i - z_j) + \beta_0 \tag{31}$$

The flow in $x$-direction and $y$-direction can be calculated using the following equations:

$$Q_x = \delta_x(z_i - z_j) + \beta_x \tag{32}$$

$$Q_y = \delta_y(z_k - z_j) + \beta_0 \tag{33}$$

The continuity equation can be discretized in the following format:

$$\frac{z_j - z_j^0}{\Delta t} + \frac{\Delta q_x}{\Delta x} + \frac{\Delta q_y}{\Delta y} = q \tag{34}$$

which is simplified to the following format:

$$\sum Q_i = A \frac{z_j - z_j^0}{\Delta t} \tag{35}$$

where $A$ is the area of cell $j$, and $\sum Q_i$ is the sum of inflow to cell $j$.

*2.3. Hydraulic Engineering Structure's HFU*

The hydraulic structures in DF-RMS such as weirs that belong to the runoff concentration unit. The weirs are important hydraulic structures used to control water resource distribution, urban flood inundation relief, and water landscape maintenance. The flow through a weir could be calculated based on the weir equation. Two types of flow were simulated: free outfall and submerged outfall weirs. The free outfall can be calculated using the following equation:

$$Q_f = mB\sqrt{2g}h_u^{1.5} \tag{36}$$

where $m$ is the free outfall coefficient, which is between 0.325 and 0.385; $B$ (m) is the weir width; $h_u$ (m) is the water height over the crest of the upstream node for the weir.

The submerged weir can be calculated using the following equation:

$$Q_s = \varphi_m B h_d \sqrt{2g(Z_u - Z_d)} \tag{37}$$

where $\varphi_m$ is the submerged weir coefficient, which is between 1.00 and 1.18; $h_d$ (m) is the water height of the downstream node for the weir; $Z_u$ (m) and $Z_d$ (m) are the water surface

elevation of the upstream and downstream node for the weir, respectively; $g$ (m$^2$/s) is the gravity acceleration coefficient, which equals 9.81.

Equations (36) and (37) can be discretized into the following format:

$$Q_f = \delta_f Z_u + \beta_f \text{ and } Q_s = \delta_s(Z_u - Z_d) \tag{38}$$

where $Q_f$ (m$^3$/s) and $Q_s$ (m$^3$/s) are the flow through the weir at the free outfall and submerge conditions, respectively; $\delta_f$, $\beta_f$, and $\delta_s$ are discretization coefficients that can be solved with the known boundary condition by the iteration method.

### 2.4. Case Test of DF-RMS

#### 2.4.1. Basic Information on the Study Area

The study area was located in the middle and downstream of the Huai River Plain, which covers the rivers and lakes from Bengbu Gate in Bengbu City to the main outlet of Hongze Lake (Sanhezha Gate) (Figure 7a). Along the mainstream of Huai River, there are eight flood plain areas (Figure 7b). There are more than 380 polder areas along the flood detention polder around Hongze Lake. Hongze Lake contains 158,000 square km of incoming water from the upper and middle reaches of the Huai River. It is one of the four largest freshwater lakes in China, which is a comprehensive plain lake integrating flood control, irrigation, shipping, water supply, power generation, and aquaculture. During the normal water period, the water flows mainly in rivers and lakes. During the flood period, as the water level rises when the upstream flow exceeds the discharge capacity of the river channel, measures such as flood diversion and discharge are used to make the water flow through gates and weirs to the flood plain. Therefore, we not only need to simulate the flood movement in the river but also the flood movement in the basin to design a flood control plan, optimize the utilization of flood control structures, and reduce flood risk.

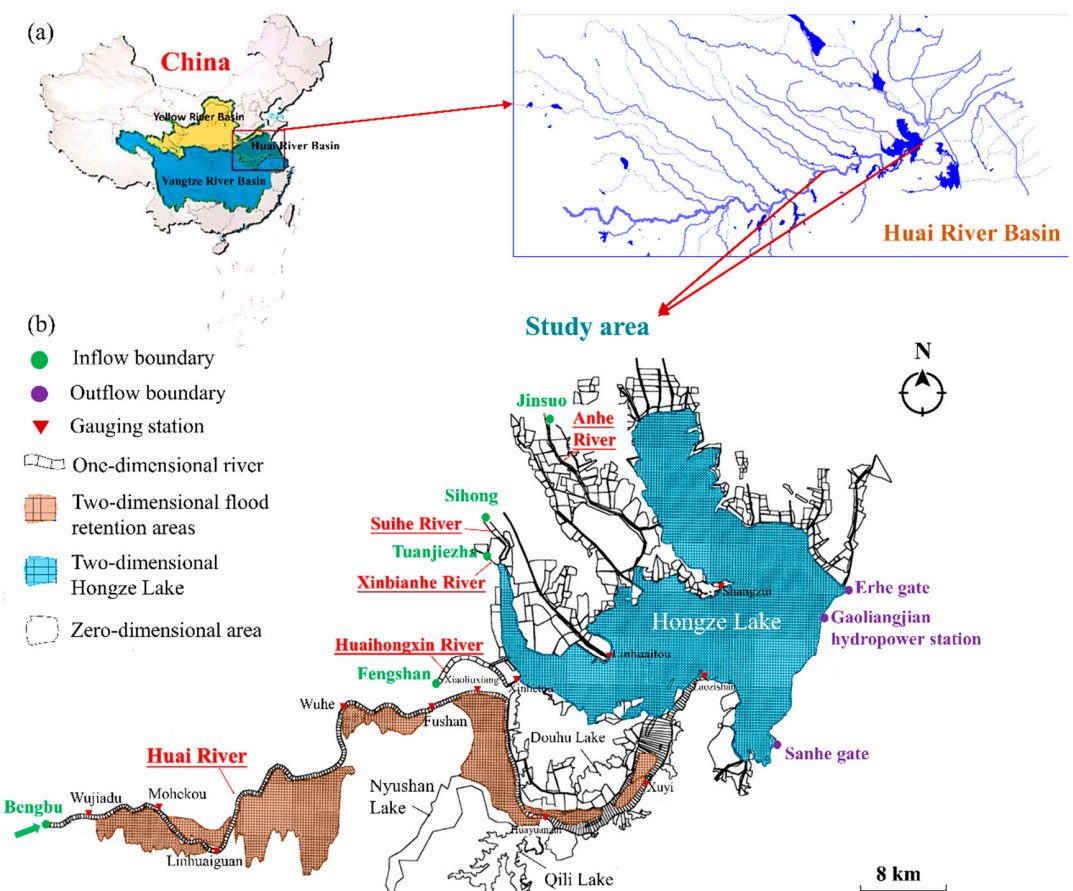

**Figure 7.** (**a**) Schematic of the study area and (**b**) boundary condition sites and generalized information.

The main inflow boundary sites of the study area include Bengbu station on the mainstream of Huai River, Fengshan station on Huaihongxin River, Jinsuo station on Anhe River, Sihong station on Suihe River, and Tuanjie gate on Xinbianhe River (Figure 7b). The main outflow boundary sites include Sanhe Gate, Erhe Gate, and Gaoliangjian hydropower station. These inflow and outflow stations with complete hydrological data provide reliable boundary conditions for the hydrodynamic simulation model. The interaction flow boundary conditions among different river basins were simulated by a hydrological model based on rainfall data. For the hydrologic model, it was simulated through hilly sub-watershed HFUs and hilly river HFUs that belong to the second paper in this series. In this case study, the hydrologic simulation was not introduced. The hydrodynamic model includes a zero-dimensional simulation of the polder area, a one-dimensional simulation of the river, a two-dimensional simulation of the Hongze Lake and flood retention areas as well as the coupling for all simulation components. The movement of floods in the basin was simulated based on proper basin generalization, appropriate numerical methods adopted for specific water flow conditions, and suitable coupling among different simulation modules.

### 2.4.2. Model Conceptualization for the Study Area

The study area was conceptualized into a specific model according to the data of the basin. Twenty-seven river channels were generalized along Hongze Lake, which were simulated using the one-dimensional component described in Section 2.1.2 (Figure 7b). The cross-section data of Huaihe River's mainstream were available, simulated using a one-dimensional algorithm. Nyushan Lake, Qili Lake, Douhu Lake, and 380 polder areas along Hongze Lake were important to the storage volume and water level changes; therefore, a zero-dimensional component described in Section 2.2.1 was used to simulate these polders. There are eight large flood retention areas along the Huaihe River's mainstream: Fangqiu, Linbei, Huayuan, Xiangfu, Pancunwa, Yaotan, Hatan, and Chenggenwei. These eight flood retention areas were considered as two-dimensional simulation areas (described in Section 2.2.2) so that using these areas to control flooding is accurately simulated/mimicked. The cell size used for the two-dimensional simulation was 500 × 500 m. Fangqiu, Linbei, Huayuan, Xiangfu, Pancunwa, Yaotan, Hatan, and Chenggenwei were divided into 326, 111, 920, 184, 570, 63, 44, and 52 cells, respectively (Table 1). Each two-dimensional simulation zone was connected to Huai River through an artificial wide weir-type connection that determined the flow interaction. Hongze Lake is a typical two-dimensional area. The lake is wide, with a 1500 km$^2$ storage area. The effects of wind stress should be considered to establish the full two-dimensional simulation model. The Hongze unit was divided into a total of 6174 computing cells with a cell size of 500 m.

**Table 1.** Model discretization of the simulation domain.

| Name | X-Resolution (m) | Y-Resolution (m) | Number of Simulation Node |
|---|---|---|---|
| Fangqiu Lake | 500 | 500 | 326 |
| Linbei section | 500 | 500 | 111 |
| Huayuan Lake | 500 | 500 | 920 |
| Xiangfu section | 500 | 500 | 184 |
| Pancunwa | 500 | 500 | 570 |
| Yantan | 500 | 500 | 63 |
| Hatan | 500 | 500 | 44 |
| Chenggenwei | 500 | 500 | 52 |
| Hongze Lake | 500 | 500 | 6174 |
| Total | - | - | 8444 |

Overall, the model included 582 cross-sections, 428 connections, and 9026 flood computation nodes that were developed for the study area. In this model, the simulation domain was generalized into 384 zero-dimensional simulation nodes, 582 one-dimensional

simulation nodes, and 8444 two-dimensional simulation nodes. The simulation domain boundary conditions were Bengbu station, Fengshan station, Jinsuo station, Sihong, and Tuanjianzha with observed flow data as the boundary conditions. The rainfall–runoff process in the simulation domain was simulated by the hydrological model and added to the corresponding unit. The observed flow data of Erhe Gate and Gaoliangjian hydropower station, such as the outlet at Hongze Lake, were adopted as the outflow boundary condition. The observed water level data of the Sanhe gate were used as the boundary condition.

## 3. Results and Discussion

The model was calibrated with the observed hydrological data of the year 1982 and validated with the observed hydrological data of the year 1991. The roughness of the Huaihe River was selected: 0.0185–0.0235 for the main channel and 0.035–0.040 for the flood plain. The roughness of the river channel around Hongze Lake was 0.025–0.035, and the roughness of the lake area was 0.015–0.065. The validation and calibration results are shown in Table 2.

**Table 2.** Calibration and validation results of the DF-RMS model.

| Location | Station Name | $Z_{psi}$ (m) | $Z_{pob}$ (m) | $R^2$ | $\Delta Z_p$ (m) |
|---|---|---|---|---|---|
| Input boundary | Bengbu | 21.49 [1] (22.11) [2] | 21.47 (22.2) | 0.998 (0.997) | 0.02 (−0.09) |
| | Sihong | 16.05 (15.19) | 16.05 (15.11) | −(−) | 0 (0.08) |
| | Tuanjiezha | 18.79 (16.57) | 18.75 (16.67) | 0.955 (−) | 0.04 (−0.10) |
| Internal part | Wujiadu | 21.13 (21.75) | 21.12 (21.81) | 0.996 (0.997) | 0.01 (−0.06) |
| | Mohekou | 20.46 (21.08) | 20.42 (−) | 0.996 (−) | 0.04 (−) |
| | Linhuaiguan | 20.05 (20.68) | 20.01 (20.64) | 0.995 (0.995) | 0.04 (0.04) |
| | Wuhe | 18.29 (18.93) | 18.31 (18.88) | 0.994 (0.994) | −0.02 (0.05) |
| | Fushan | 17.4 (18.07) | 17.39 (17.98) | 0.993 (0.995) | 0.01 (0.09) |
| | Xiaoliuxiang | 17.13 (17.79) | 17.13 (17.74) | 0.994 (0.995) | 0 (0.05) |
| | Huayuanzui | 15.75 (16.60) | 15.73 (16.64) | 0.991 (0.987) | 0.02 (−0.04) |
| | Xuyi | 14.58 (15.38) | 14.63 (15.38) | 0.977 (0.99) | −0.05 (0) |
| | Xinhetou | 13.36 (14.42) | 13.74 (14.42) | 0.67 (0.81) | −0.38 (0) |
| | Laozishan | 13.31 (14.18) | 13.29 (14.23) | 0.915 (0.981) | 0.02 (−0.05) |
| | Linhuaitou | 12.91 (14.08) | 12.93 (14.04) | 0.935 (0.984) | −0.02 (0.04) |
| | Shangzui | 12.88 (14.05) | 12.74 (13.95) | 0.906 (0.947) | 0.14 (0.10) |
| Output boundary | Gaoliangjian | 12.88 (14.06) | 12.77 (14.02) | 0.944 (0.981) | 0.11 (0.04) |

[1] The calibration results for the year 1982; [2] the validation results for the year 1991. $Z_{psi}$ (m) is the simulated peak water elevation, $Z_{pob}$ (m) is the observed peak water elevation, $R^2$ is the determination coefficient of simulated and observed results, $\Delta Z_p$ (m) is the difference between the simulated and observed water elevation, $\Delta Z_p = Z_{psi} - Z_{pob}$.

Figure 8a,b gives the calibration results of the case for the year 1982, and Figure 8c,d gives the validation results of the case for the year 1991. The solid line in the figure is the simulated result, and the dotted line is the actual observed data. Table 2 shows the determination coefficients, which reflect the agreement between the observed data and simulated results of each station.

Table 2 shows that the validation results for 1991 match the observed data well. The difference between the simulated and observed peak water surface elevation for all stations was within 10 cm, and the determination coefficient was above 0.94 (except for 0.81 for Xinhetou). The validation results for the year 1982 showed that the observed and simulated peak water surface elevation of all stations were in good agreement, with a determination coefficient above 0.915 (except 0.67 and 0.906 for Xinhetou and Shangzui stations).

The difference in observed and simulated water surface elevations with calibrated and validated model for Xinhetou was larger than for the other stations because Xinhetou is directly linked downstream of Hongze Lake. A two-dimensional model with a cell size of 500 m was used to simulate Hongze Lake, which is a very shallow lake. The cell size was relatively large, and the lake is shallow in the inlet, so it was difficult to simulate the flow of the Xiaohetou inlet. The two-dimensional simulation can reflect the actual inlet flow when the lake water is deep. In 1991, the water surface elevation for the lake was

deep, and the simulation results of the water surface elevation at the Xinhetou were better than the results for the year 1982 (Figure 9).

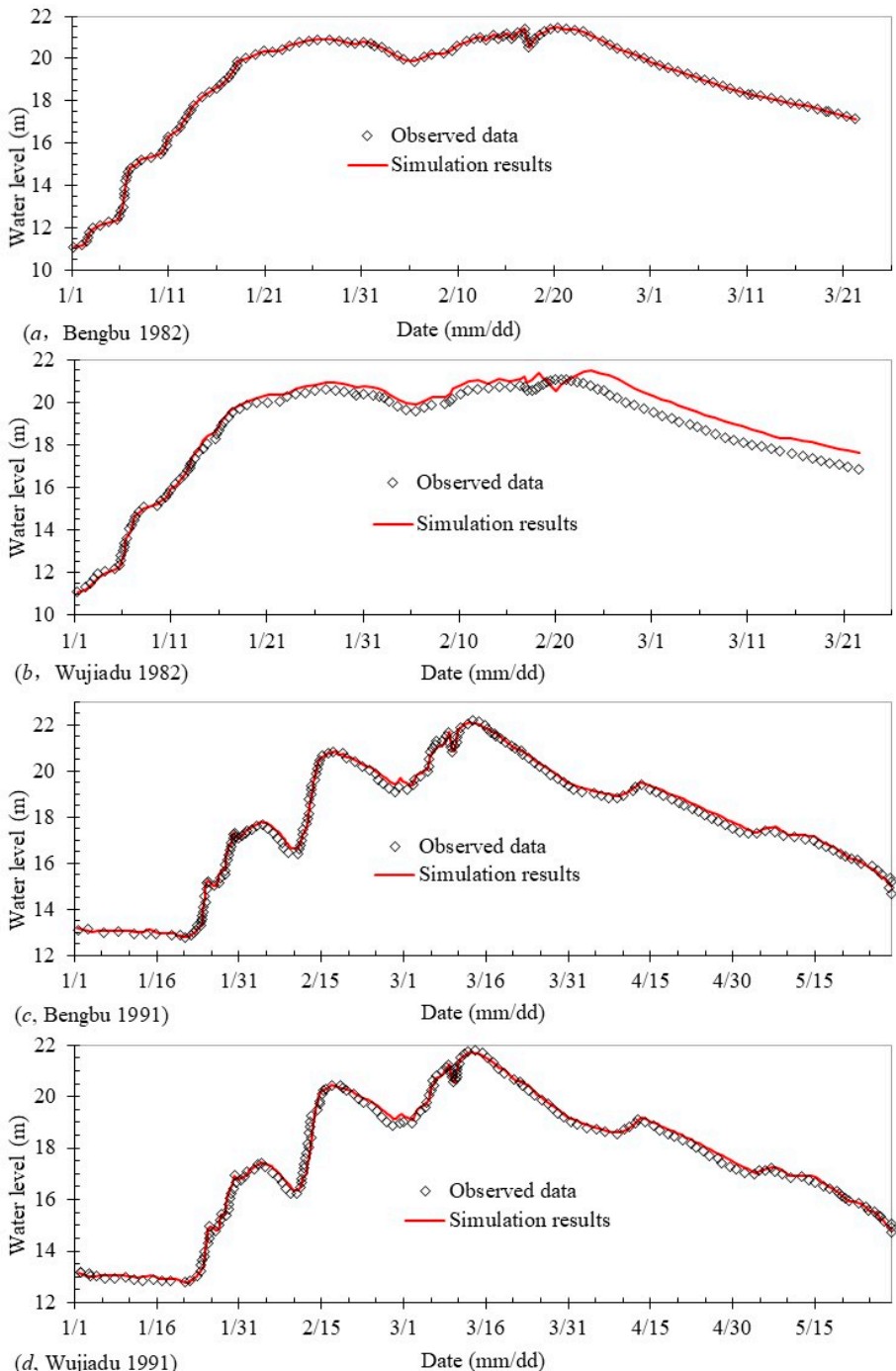

**Figure 8.** Simulated and observed water surface elevations with calibrated and validated model of two stations in 1982 and 1991: (**a**,**c**) for Bengbu station; (**b**,**d**) for Wujiadu station.

The results for the year 1982 and 1991 show that the calibrated and validated model can simulate the flow movement in the river basin and correctly reflect the rainfall-runoff process of the basin. The flow field distribution of Hongze Lake on 03/01/1991 is also shown in Figure 10. The two large inflows from the Huai River and Huaihongxin River led to large flow momentum. The main outflow at Sanhe Gate and Gaoliangjian hydropower station show large velocities. Overall, it predicted the flow field distribution under actual conditions reasonably well.

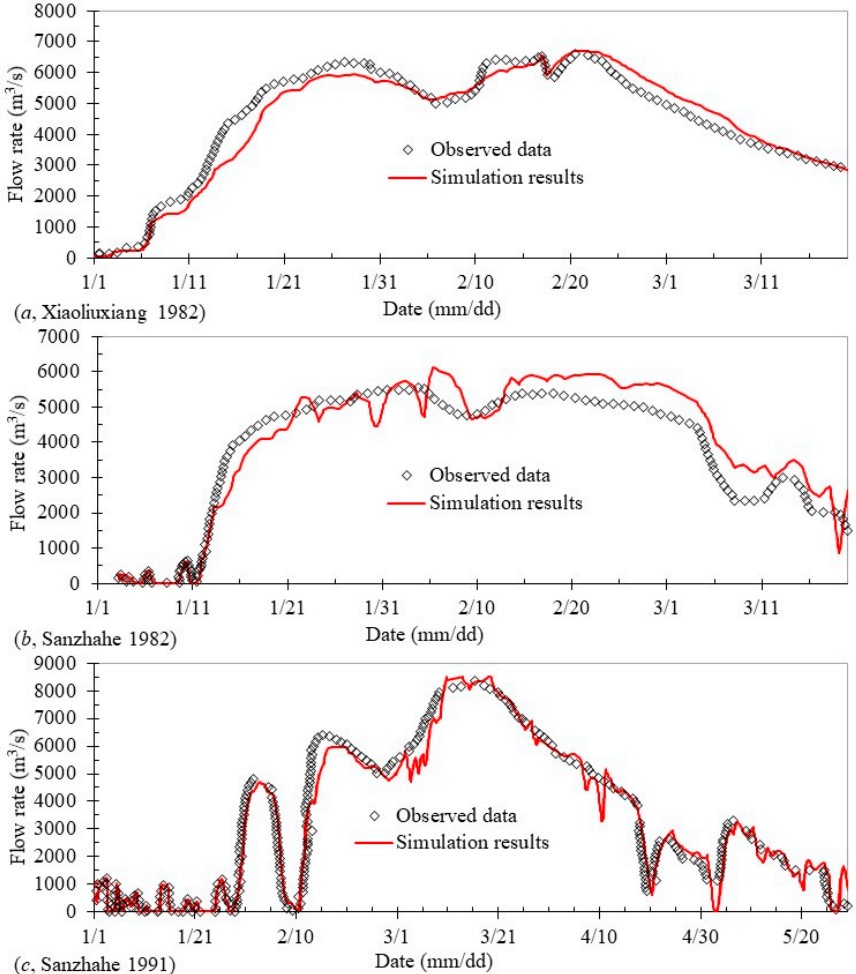

**Figure 9.** Simulated and observed flow rates with calibrated and validated model of two stations in 1982 and 1991: (**a**) for Xiaoliuxiang; (**b**,**c**) for Sanhe Gate station.

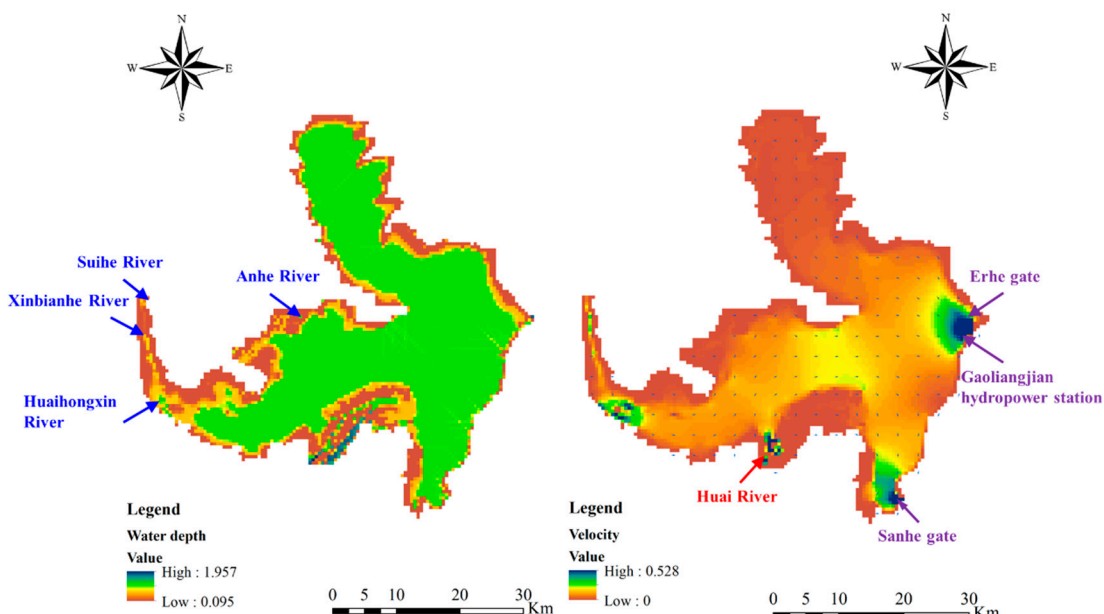

**Figure 10.** Simulated flow depth (**left**) and velocity (**right**) distributions of Hongze Lake.

## 4. Summary and Conclusions

The numerical procedures that DF-RMS uses to model the hydraulic processes in runoff concentration, flow movement in river networks, and lakes are discussed in detail in this paper. The river networks were generalized into different types: one-dimensional river branches, dendritic river networks, loop river networks, and intersecting river networks. The matrix used for flow movement simulation for different kinds of river network components was derived step by step. The matrix used to simulate flow in river network intersections was also obtained with the consideration of two-dimensional flow in river network intersections. The flow in lakes was treated as either a zero-dimensional ponding node or two-dimensional flow movement simulated by the operator-split algorithm. One testing case with different kinds of river networks and lakes was simulated with DF-RMS and calibrated, validated by observed data. The simulation results of DF-RMS showed great consistency with the observed data, which proved that DF-RMS is a reliable program to simulate the flow movement in complex river networks and lakes. This is a direct contribution to modeling hydrologic/hydraulic response from a non-homogenous catchment. It could be applied to flood control planning scheme simulation and design.

**Supplementary Materials:** The detailed process of river intersection flow simulation matrix buildup, loop river flow simulation matrix buildup, and part of the hydrologic simulation process about the hydraulic modeling system in DFBMS are available online at https://www.mdpi.com/2073-4441/13/5/649/s1.

**Author Contributions:** The work was conducted by X.L., C.W., G.C., X.F., P.Z., and W.H.; this paper was first written by X.L., G.C., and C.W. reviewed and improved the manuscript with comments; the data compilation and statistical analyses were completed by all authors. All authors have read and agreed to the published version of the manuscript.

**Funding:** This research was financially supported by the National Key Research and Development Program of China (2018YFC1508200), the Fundamental Research Funds for the Central Universities (B200202029 and B200202030), and Project 41901020 supported by NSFC, and Hydraulic Science and Technology Program of Jiangsu Province (2020003).

**Institutional Review Board Statement:** Not applicable.

**Informed Consent Statement:** Not applicable.

**Data Availability Statement:** The data presented in this study are available on request from the corresponding author.

**Conflicts of Interest:** The authors declare no conflict of interest.

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
