# Peer review of "Distributed-Framework Basin Modeling System: Ⅲ. Hydraulic Modeling System"

_water, doi:10.3390/w13050649_

Round 1
Reviewer 1 Report
The revision looks good and my comments have been resolved. Good work.
Author Response
Thanks again for your comments. We appreciate your work and help with our paper.
Reviewer 2 Report
I truly believe that the 4 papers could be synthesized into just one single paper, but I’ll leave that decision up to the Editor. Having said this, the authors have addressed each of my concerns and they are applauded for improving the papers. I appreciate the contribution of the authors in reporting their hydrologic modeling framework and I encourage them to work in future implementations with public repositories with codes and tutorials written in English for easier access to the international community.
Author Response
Thanks again for your comments. We appreciate your work and help with our paper. We negotiate with the editors and get the permission from the editors to publish these four papers after major revisions. We are also working on the code publication and English tutorials now. The DFBMS has been developed for more than 30 years, we do need some time and effort to transfer the complicated system and tutorials to English version.
Reviewer 3 Report
The authors of the manuscript “Distributed-Framework Basin Modeling System: Hydraulic Modeling System (Ⅲ)” revised their manuscript according to the reviewers’ comments. The new output is an ameliorated version of the first manuscript. I recommend the minor revision of the manuscript, before being accepted for publication, based on the following comments:
Comment 1. The authors should have an introductory small paragraph talking about the hydraulic modelling. For example, I would propose the below paragraph. Thereafter the manuscript will be very well introduced.
“Floods are extreme phenomena that need to be accurately assessed for the protection of mankind activities. Hence, mathematical tools focused on hydraulic simulations are designated as the most holistic approach to describe – simulate the flood events and determine the flood hazard (Reil at al., 2018). Hydraulic modelling includes one dimensional (1D), two-dimensional (2D), or three dimensional (3D) approaches, regarding the number of dimensions in which the flow path is solved, as well as non-numerical models (Costabile and Macchione, 2015). The current coupling of hydraulic models with geographic information systems (GIS) have facilitate the development and application of hydraulic models (Zerger and Wealands, 2004). The use of remote sensing products, such as Digital Elevation Models (DEMs), in hydraulic modeling is also a modern commonly used approach (Smith 1997).
Where:
- Reil, A., Skoulikaris, C., Alexandridis, T. K., & Roub, R. Evaluation of riverbed representation methods for one‐dimensional flood hydraulics model. Journal of Flood Risk Management, 2018, 11.2: 169-179.
- Costabile, P. & Macchione, F. Enhancing river model set-up for 2-D dynamic flood modelling. Environ Model Software, 2015, 67, 89–107.
- Zerger, A. & Wealands, S. Beyond modelling: linking models with GIS for flood risk management. Nat Hazards, 2004, 33, 191–208.
- Smith L.C. Satellite remote sensing of river inundation area, stage and discharge: a review. Hydrol Process, 1997, 11, 1427–1439.
Author Response
Thanks again for your comments. We appreciate your work and help with our paper. We think the comment is quite useful and helpful for our manuscript. In this case, we have updated the introduction part and added references based on the reviewer’s comments.
This manuscript is a resubmission of an earlier submission. The following is a list of the peer review reports and author responses from that submission.
Round 1
Reviewer 1 Report
General comments
As a continuation from the 2nd manuscript, the authors within this 3rd manuscript demonstrate information about the hydraulic component of the proposed DFBMS, namely the distributed-framework river modeling system (DF-RMS). The theoretical part and its connection to the various HFUs, i.e. Plain river HFU, Lakes and Reservoirs’ HFU and Hydraulic engineering structures’ HFU is well documented. The case study area is also well described. The outputs are also well presented. Overall, this 3rd manuscript is a well-structured manuscript which presents the new methods proposed by the authors and requires minor revisions.
Specific comments
The introduction part is lacking of a last paragraph mentioning the aim of the specific manuscript and the proposed advantages.
Line 32. References 1 and 2 cannot be evaluated since they are in Chinese. The same with the references 3, 4, 5.
Lines 77-79. “The runoff on the underlying surface under grid processing was assigned to the nearest river cross section individually rather than concentrated to the subwatershed outlet or the upstream entrance.” This is a statement that is not given in the 2nd manuscript. It should be added in the 2nd manuscript with the proper details.
Reviewer 2 Report
This series of 4 papers presents the description of a distributed basin modeling system composed of several components. The strength of the model laid on the flexibility and number of processes that can be integrated and modeled across the hydrologic and hydraulic components. In general, the papers are well-written, and the methods included in the hydrologic and hydraulic components are well presented. Having said this, I do not consider proper to present this work as a series of 4 papers, the overall structure looks more suitable for a dissertation document or a chapter book, but the presented format does not fit with the overall goal of a scientific paper, in which should maximize the synthetizes of methods, results, and discussions without losing accuracy and transparency, which reminds me the popular said: “I didn't have time to write a paper, so I wrote a book.” I encourage the authors to reconsider to condense the work into one single paper in order to show their wonderful work. Below, I’m describing my major and minor comments for all 4 papers.
[Major Comments]
- The 4 papers should be presented as a single paper. “Paper 1” can be easily synthesized as the introduction section, “Paper 2” and “Paper 3” is the section method, and “Paper 4” would be the Case Study. I noted through the papers several redundancies that can be avoided in order to achieve the best synthetizes in the work. For instance, “Paper 2” and “Paper 3” show a Case study; however, that should be the main purpose of “Paper 4”. There are several sections in “Paper 2” and “Paper 3” that can be moved to a Supplemental Material section (See Minor Comments).
- As I mentioned before, the strength of the study is found in the hydrologic and hydraulic components. However, “Paper 4” decreases the overall impact of the presented model. The authors were limited to show that the proposed model was able to replicate the discharge, water depth in some gauges just for a short period of time (calibration 1 year, validation 1 year). In general, there is not an analysis of the spatial distribution of the model performance, and there is no understanding of how the different model components, either hydrologic or hydraulic, is improving the representation of the hydrologic processes. This paper shows a new hydrologic model framework, therefore, should be expected to find an extended analysis of the different capabilities of the models showing the improvement of the model with and without different components
- The authors did not provide the source code or repository of the model. This is essential for future implementations of the model in the hydrologic community. In the case that the model is not available to the public, the authors must provide further details about the configuration in computational times used to run the simulations.
Minor Comments on: “Distributed-Framework Basin Modeling System: Overview and Model Coupling (I)”
[Line 25] What advantages? This statement is ambiguous
[Line 33] What does FH69 stand for?
[Line 106] Be careful using the argument of “Temporal GIS”, this is a matter of perspective, somebody could argue that including time-series to represent rainfall fields is sufficient to be in the realm of “Temporal GIS”. Besides, geological models do not seem an appropriate example to show the inconvenience of temporal representations in the current hydrological models, note that the geological processes evolve in a dramatic larger time scale in comparison to hydrologic processes explored in most of the hydrologic models.
[Figure 5] The captions need to be improved
Minor Comments on: “Distributed-Framework Basin Modeling System: Hydrologic Modeling System (II)”
[Line 20] Only two HFUs? what about the other 9 HFUs? Is there any model documentation?
[Line 122] What do SFD and MFD stand for?
[Section 2.1 2.1.1, 2.1.2] This section could be omitted or summarized in one or two paragraphs. Most of this content may be considered as general knowledge in the hydrologic community (e.g. estimation of flow direction D8), so there is no need to be so explicit in its development. Another option is included as Supplemental Material.
[Equation 2] What water depth and Chezy coefficient are used in Eq 2? Are they varying over space? Or is it just the DEM elevation used as the water depth in this case? Or are assumed constant across all the DEM processing?
[Lines 261-268] The runoff generation process and the overland flow must be explained in this section! The authors are just limited to provide some references; this is one of the key elements in the description of any hydrologic model.
[Lines 270-278] Be specific with the hydrograph method. This statement is vague, what equations and approximations are the authors using for the hydrograph routing method?
[Section 2.2.3] So does the plain overland runoff generation considers land use, but the Hilly- subwatersheds do not? Section 2.2.3 is nicely documented, however, section 2.2.1 is poorly described.
[Line 320] What specific parameter range? Be specific.
[Line 323] How necessary is to include this complexity in modeling runoff on the paddy fields? Have the authors provided any evidence of the adequacy in including this process? This should be explored in high detail in “Paper 4”
[Line 361] Provide ranges for Hp, Hu, and Hd
[Section 2.3] What about the modeling in woodland land use?
[Section 2.2.4] Include a description of the overland runoff method used. Again, what about the woodland surface?
[Section 2.3] This is not necessary if it is mentioned in “Paper 3”
[3 Study Case] This should be part of “Paper 4”
[Section 3.2] The evolution of the model performance needs to be improved. Please consider using the Nash–Sutcliffe model efficiency coefficient since has been used as a standard in the hydrologic community
[Section 3.2] Be specific in how the calibration was performed. What method? And what parameters were calibrated in this case study?
[Section 3.2] What was the computational time?
Minor Comments on: “Distributed-Framework Basin Modeling System: Hydraulic Modeling System (III)”
[Section 2] Large part of this paper could be included as an Appendix or Supplementals Information
[General] The equation numbering is incorrect, please verify.
[Lines 42-59] If there is no further discussion about these aspects through the paper, then this section should be removed.
[Section 2.4] This should be part of the “Paper 4”
[Line 443-Line 444] Rewrite “1982 cases...” for “case for the year 1982”; same for 1991.
[Figure 10] It is a better option to use a color bar to show the velocity field
Minor Comments on: “Distributed-Framework Basin Modeling System: Application in Taihu Basin (IV)”
[Figure 11] Are there only 4 streamflow gauges? Show statistics RMSE, Nash–Sutcliffe model efficiency
[Section 3.2] Why is the validation period and calibration period so short? I assume that there should be longer streamflow records within the basin, however, the authors only used one year for calibration and one for validation which obscures the true overall model performance that could be achieved with larger hydrologic records.
[General] Show the drainage area associated with each streamflow station
Reviewer 3 Report
This third paper describes the methodology of DF-RMS for hydraulic modeling. A case study is shown to demonstrate its reliability in simulating flow when compared with observation data. I have some concerns about the results analysis. Please see my comments below:
1. Since this manuscript has a lot of equations (which is fine), can you make sure all of them having same formats and layouts? I found a lot of indentation is different.
2. In Figure (b), I think you meant "inflow boundary" and "outflow boundary" instead of "input/output boundary"?
3. Line 425, Coriolis force should be considered for large-scale simulation, but for the size of this area, is it really needed? I'm curious about this (not saying you should not consider Coriolis).
4. Can you clarify how you did calibration and validation separately? Was one year data used for each purpose separately or did you combine them together?
5. This figure needs improvements. You can actually plot velocity magnitude in a much cleaner way by using colored map and incorporate direction information as another layer.
6. Since you show the spatial distribution of flow, have you also compared modeled velocity with any observation? What would be a good evidence that your model simulates flow well?